

# Design of smart citrus picking model based on Mask RCNN and adaptive threshold segmentation

Ziwei Guo[1], Yuanwu Shi[2] and Ibrar Ahmad[3]

[1] Hubei University of Technology, Wuhan, China
[2] Wuhan Textile University, Wuhan, China
[3] Department of Computer Science, University of Peshawar, Peshawar, Pakistan

## ABSTRACT

Smart agriculture is steadily progressing towards automation and heightened efficacy. The rapid ascent of deep learning technology provides a robust foundation for this trajectory. Leveraging computer vision and the depths of deep learning techniques enables real-time monitoring and management within agriculture, facilitating swift detection of plant growth and autonomous assessment of ripeness. In response to the demands of smart agriculture, this exposition delves into automated citrus harvesting, presenting an ATT-MRCNN target detection model that seamlessly integrates channel attention and spatial attention mechanisms for discerning and identifying citrus images. This framework commences by subjecting diverse citrus image classifications to Mask Region-based CNN's (Mask RCNN's) discerning scrutiny, enhancing the model's efficacy through the incorporation of attention mechanisms. During the model's training phase, transfer learning is utilized to expand data performance and optimize training efficiency, culminating in parameter initialization. Empirical results notably demonstrate that this method achieves a recognition rate surpassing the 95% threshold across the three sensory recognition tasks. This provides invaluable algorithmic support and essential guidance for the imminent era of intelligent harvesting.

# INTRODUCTION

In an era marked by the relentless escalation of labor costs, an increasing number of enterprises are turning to ever-advancing science and technology as they steadily phase out antiquated, labor-intensive production methods. The discourse of intelligence is becoming an omnipresent refrain within corporate corridors. Presently, computer vision technology has found its foothold in the realm of industrial production, owing to the burgeoning demand for classification, detection, and localization along the industrial production lines. As the industry undergoes ceaseless evolution, manual classification and detection have become increasingly inadequate in meeting the industry's burgeoning demands for durability and efficiency. High-precision computer vision technology stands as the pivotal impetus propelling industrial intelligence to the forefront. In the sphere of

Corresponding author
Ziwei Guo, 102111404@hbut.edu.cn

agricultural production, concomitant with the maturation of artificial intelligence and deep learning technology, the purview of intelligent farm management, the monitoring of crop diseases through the prism of computer vision, the surveillance of ripening stages, and the stringent quality control have emerged as focal points, catalyzing an upsurge in agricultural production efficiency (*Giakoumoglou, Pechlivani & Tzovaras, 2023*).

In natural settings, conventional algorithms for fruit target detection traditionally rely on target characteristics such as color, shape, texture, and other distinctive attributes to facilitate the classification, detection, and segmentation of target images (*Tang et al., 2023*). However, as the integration of big data applications deepens and artificial intelligence undergoes rapid evolution, deep learning technology is increasingly making its presence felt in the domains of agriculture and computer vision.

In the realm of traditional machine learning techniques, target detection predominantly hinges on meticulously handcrafted feature extraction and the application of machine learning algorithms. These methods encompass the utilization of Haar cascade detectors, coupled with HOG features employing SVM classifiers, among others (*Zhao et al., 2019*). While these approaches still retain their efficacy in specific scenarios, they grapple with limitations when confronted with intricate backgrounds, variations in scale, and instances of target occlusion. In light of the swift progression of deep learning technology, computer vision-based research methods have assumed a prominent role in target detection. Target detectors can be broadly categorized into two distinct types: one-stage and two-stage, differing significantly in their detection methodology and processes. A noteworthy exemplar of a single-stage detector is the YOLO (You Only Look Once) method, renowned for its real-time target detection capabilities, directly outputting bounding box and category information by framing target detection as a regression challenge (*Tian et al., 2019*). Although YOLO excels in speed, it faces certain challenges when it comes to detecting diminutive targets and instances of target occlusion. Single-stage detectors, known for their swiftness (*Xu et al., 2023*), are apt for real-time applications, albeit potentially marginally less precise than their two-stage counterparts. Two-stage detectors, on the other hand, typically encompass methodologies such as Faster R-CNN and Mask Region-based CNN (Mask R-CNN) (*Girshick, 2015*). These approaches introduce an intermediate region proposal step to enhance accuracy in the target detection process, and Mask R-CNN further excels in instance segmentation. Thus, the incorporation of these widely embraced, high-performance data modeling techniques assumes paramount significance in the advancement of agricultural harvesting practices (*He et al., 2017*).

Within the pages of this scholarly work, we embark on an exploration of the intricacies surrounding the intelligent recognition quandary in the context of fruit harvesting, a pivotal facet of the burgeoning field of intelligent agriculture. In response to this challenge, we put forth a cutting-edge Mask RCNN-based intelligent detection model, meticulously tailored for the purpose of citrus picking. Our endeavor is driven by the aspiration to furnish an innovative benchmark for the realization of intelligent and fully automated agricultural practices. The particular contributions of this paper are elucidated as follows:

(1) Employing the innovative DONI model, we conducted targeted data extraction and model pre-training on both the COCO and VOC datasets, customizing the process to meet

the specific demands and characteristics of target recognition. This approach contributes to the advancement of precision and adaptability in target identification.

(2) Pioneering the establishment of a comprehensive citrus image database featuring three distinct categories, including the challenging ''unobstructed'' category, we subsequently tailored our model training to this specialized dataset. Further enhancing our methodology, we carried out reinforcement training based on the solid foundation laid through transfer learning. This initiative significantly contributes to the development of robust models geared towards real-world citrus harvesting scenarios.

(3) Introducing the cutting-edge ATT-MRCNN citrus target detection model, an extension of the Mask RCNN architecture, specifically designed to achieve unparalleled accuracy in detecting citrus targets across diverse natural conditions. This model stands out as a crucial technological advancement, playing a pivotal role in the intelligent evolution of citrus harvesting processes. In the reminder of this article, related work is described in 'Related Works'. 'Methodlogy' established the ATT-MRCNN model. Experiment results and related analysis is detailed illustrated in 'Experiment' and 'Discussion' discusses research significance. The conclusion is drawn at the end.

## RELATED WORKS

For the intelligent harvesting of citrus, it is necessary to first monitor its objectives and consider the working conditions under different weather and working environments in order to achieve its practical application. Therefore, this article provides a detailed introduction to the research on object detection and its current application status in agriculture.

### Research in object detection

Target detection stands as a cornerstone within the expansive domain of computer vision and pattern recognition, facilitating the identification and precise localization of objects of interest within images. In the era preceding the advent of deep learning methodologies, traditional target detection approaches primarily encompassed three fundamental components: the selection of regions *via* a sliding window strategy, feature extraction techniques relying on scale-invariant feature transform (SIFT) (*Lindeberg, 2012*), histogram of oriented gradient (HOG) (*Surasak et al., 2018*), and similar methodologies. However, the sliding window selection strategy employed by conventional techniques was indiscriminate, leading to excessive window redundancy and subsequently, high time complexity. Furthermore, traditional target detection methods were contingent upon manually crafted features, which often exhibited limited robustness and poor generalization. In tandem with the profound advancements in deep learning, there has been a substantial enhancement in the precision of target detection. Consequently, deep learning-based target detection algorithms have gained significant prominence, rapidly becoming the vanguard of research in the field. Notably, ResNeSt (*Zhang et al., 2020*) introduced a split-attention module, drawing inspiration from the principles of SENet (*Hu, Shen & Sun, 2018*). This module leverages channel attention to assign distinct weights to individual channels, encapsulating their relative significance. Additionally,

it incorporates elements from SKNet (*Li et al., 2019*) to infuse an attention mechanism into the convolutional kernel. EfficientDet (*Tan, Pang & Le, 2020*) innovatively introduced the BiFPN structure, an evolution of the PANet, characterized by the removal of single input nodes and the connection of edges between input and output nodes within the same hierarchy. This fosters the integration of an expanded spectrum of features, complemented by the incorporation of both "bottom-up" and "top-down" paths, which are regarded as foundational layers and are iteratively repeated. CornerNet (*Law & Deng, 2020*), on the other hand, focuses on localizing target boxes by predicting the coordinates of the top-left and bottom-right corners, thereby departing from the conventional bounding box paradigm. This approach employs two distinct branches to generate heat map, one for each corner, corresponding to the various object categories. These thriving areas of research underscore the prevailing emphasis on enhancing the backbone network and fine-tuning anchor points to refine target recognition frameworks, thus driving advancements in target detection. Nevertheless, it is worth noting that deep learning-based target detection methods may still present challenges in the context of multi-class target recognition within common numerical sets. Therefore, narrowing the scope of target detection recognition and intricate network refinement remain pivotal strategies for elevating the accuracy of recognition, especially for specific focus targets. The object detection based on deep learning technology has important application significance in citrus picking. Firstly, through deep learning algorithms, precise recognition and localization of citrus fruits can be achieved, improving the operational efficiency of harvesting robots. This technology can effectively distinguish mature citrus fruits from immature or diseased fruits, helping farmers to pick them quickly and accurately, and improving the quality and yield of agricultural production. Secondly, the object detection technology of deep learning can also help reduce labor costs and improve the automation level of the picking process. By applying machine learning models to visual perception, agricultural producers can reduce their dependence on professional pickers and achieve more intelligent and efficient citrus harvesting. This not only helps to improve the competitiveness of agricultural production, but also helps to alleviate the problem of labor shortage and provide feasible solutions for the modernization of agriculture.

## Research on fruit object detection

Traditional machine learning methodologies for fruit detection predominantly rely on the analysis of external attributes of various fruits, encompassing features such as color, shape, and texture. *Zhao et al. (2016)*, for instance, devised a model that classified fruits based on unique external features, and employed a KNN classifier to achieve an 84% detection rate on a custom-built dataset, derived from extensive external feature analysis of a diverse range of fruits. Similarly, *Chaivivatrakul & Dailey (2014)* focused on texture feature analysis to classify fruits with different textures, albeit this model necessitated the presence of distinct textures on the target object for effective detection, resulting in variations in detection rates. It is worth noting that detection for targets with subtle textures posed a particular challenge for these models. In pursuit of fruit detection, many have employed the Hough transform to fit the external shape of fruit targets within

fruit images. *Sa et al. (2016)*, for example, employed a detection model that combined image data of bell peppers with various color models, employing transfer learning to accurately detect bell peppers within greenhouse environments. *Bargoti & Underwood (2017)* introduced a fruit detection algorithm founded on multiscale multilayer perceptron and CNN feature learning. They addressed the cumbersome nature of manual feature extraction by incorporating a fusion of multilayer perceptron and CNN segmentation methods, which in turn facilitated image segmentation for target identification. Watershed segmentation and Hough transform algorithms were subsequently used for detection and counting of segmented maps, thereby enhancing model performance in segmentation and detection tasks. However, the model's generalizability across diverse fruit types and scenarios warrants further investigation. *Wan & Goudos (2020)* proposed an enhanced Faster R-CNN framework for multi-class fruit detection, which entailed the creation of a self-learning fruit image library to establish a high-resolution detection approach. This method involved the optimization of convolutional and pooling layers within the existing Faster R-CNN and dynamic parameter adjustment during training. *Chen et al. (2021)* presented a Des-YOLOv4 apple detection algorithm, aiming to enhance both the detection speed of the YOLOv4 model and reduce its rates of misclassification and omission. To achieve this, they optimized the YOLOv4 model by incorporating elements from the DenseNet network architecture. Additionally, they proposed a classification loss function based on the average accuracy loss to harmonize the training of positive and negative samples.

Within the aforementioned fruit target recognition research, it becomes evident that the simplicity of fruit targets has led to a rather uncomplicated overall model usage. The majority of these models are grounded in traditional frameworks such as Faster R-CNN, YOLO, and similar methodologies. Consequently, this article endeavors to enhance the existing common target detection methods, integrating model preprocessing techniques with data preprocessing and transfer learning. The aim is to expedite model training while simultaneously enhancing accuracy, thereby achieving a high recognition rate and precision even when working with limited sample datasets.

# METHODLOGY

In response to the current research status of object detection and fruit object detection in agriculture, this article aims to build a basic model for citrus object detection based on ResNet and Fast RCNN, and strengthen the model on this basis. The specific process is as follows:

## Convolutional neural networks and ResNet

The convolutional neural network (CNN) represents a deep learning architecture tailored for the processing and analysis of data structured in grids, particularly well-suited for image and video data. Its strength lies in the ability to extract intricate features through convolutional operations. ResNet, short for residual neural network, stands as a specialized architecture within the domain of deep CNN. It was conceived to address the issues of

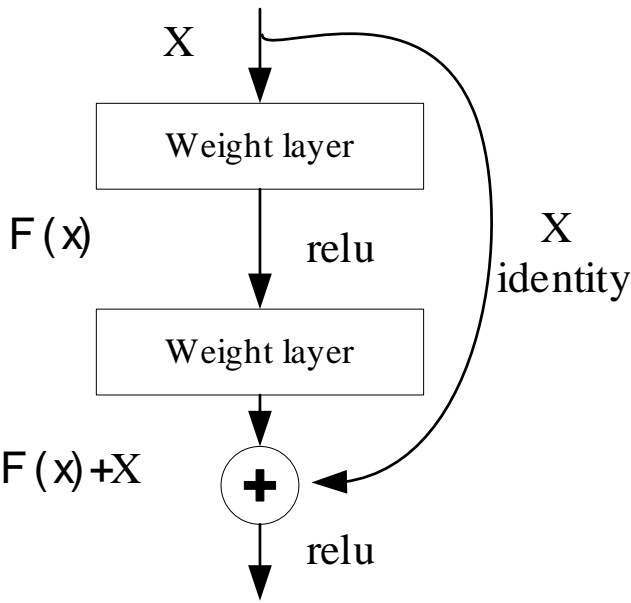

**Figure 1** The structure of ResNet.

gradient vanishing and exploding that often plague the training of deep neural networks (*Targ, Almeida & Lyman, 2016*). The core structure of ResNet is depicted in Fig. 1.

The pivotal concept underpinning ResNet revolves around the incorporation of residual blocks, as expressed in Eq. (1). These blocks enable the network to focus on learning the residual function in lieu of the original function. This design permits the network to effortlessly grasp constant mappings without deteriorating performance (*Wightman, Touvron & Jégou, 2021*). As a result, the architecture can attain considerable depth, extending to hundreds of layers, rendering ResNet highly adept at excelling in object detection.

$$F(x) = H(x) + X \tag{1}$$

where $F(x)$ is the output, x is the input, and $H(x)$ is the residual function, which represents the residuals to be learned by the network. By adding x to $H(x)$, the residual block enables the network to learn the residuals.

## Mask R-CNN model

Mask R-CNN (Mask Region-based CNN) constitutes a deep learning model that serves as an extension of the Faster R-CNN model. It distinguishes itself by producing pixel-level semantic segmentation masks for each discerned object instance. The comprehensive architecture of Mask R-CNN is illustrated in Fig. 2. It can be broadly dissected into three distinct modules: the feature extraction network, the inter-region network, and the RoI Align layer. These components collaboratively fulfill the mission of object detection and

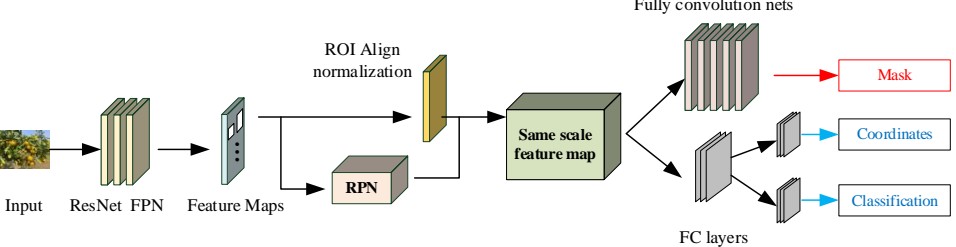

**Figure 2** **The framework for the Mask R-CNN.**

classification, ultimately culminating in the utilization of the fully connected layer and its associated loss function (*Vuola, Akram & Kannala, 2019*).

Mask R-CNN introduces a Mask network branch to enhance RoI prediction segmentation in the Faster R-CNN framework. It also presents the incorporation of bilinear interpolation within the RoIAlign layer, thus yielding more precise pixel data and consequently elevating the accuracy of the target detection branch. The Mask R-CNN model primarily comprises CNNs for shared feature layer extraction, Region Proposal Networks (RPNs) responsible for generating proposal frames, an output model that maps candidate frames into fixed-size feature maps, facilitating subsequent classification and regression operations. Furthermore, it encompasses an output model dedicated to frame regression and segmentation masks. The architecture encompasses an RPN aimed at suggesting frames, the RoIAlign layer designed to map candidate frames into uniform feature maps for subsequent classification and regression operations, and a dedicated output model for frame regression, classification, and segmentation masks. The Mask R-CNN algorithm facilitates the processing of tobacco images through a backbone feature extraction network. The resulting shared feature layer, enriched with tobacco image characteristics, is directed to the RPN, where frames of varying sizes and proportions are meticulously constructed. Subsequently, frames of interest, containing the targets, proceed to the RoIAlign layer, where they are aggregated into a standardized feature map. The role of the RoIAlign layer in object detection neural networks is to achieve fine sampling within the region of interest by using bilinear interpolation, thereby solving the spatial misalignment problem that may be introduced in the RoIPool layer. By accurately mapping the positional relationship between the input image and the output feature map, RoIAlign improves the accurate capture of target boundaries and spatial information by the object detection network, thereby improving model performance, especially in tasks with high requirements for target localization accuracy. Lastly, the object features, obtained through pooling within the RoIAlign layer, are channeled into fully connected layers and a fully convolutional network for the purposes of object classification and instance segmentation. This comprehensive process yields the citrus target detection results, integral to the study presented in this paper, facilitating the recognition of citrus data across diverse scenarios.

In the Mask R-CNN output module, the loss sums for classification, regression, and prediction masks are computed using a multi-task loss function for each RoI produced by

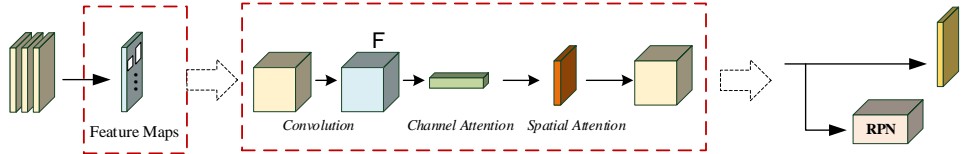

**Figure 3** Model enhancement by channel spatial attention mechanism.

the RoIAlign layer output.

$$L = L_{cls} + L_{box} + L_{\text{mask}} \tag{2}$$

where $L_{cls}$ is the classification loss, and $L_{box}$ is the box regression loss: and $L_{\text{mask}}$ is the segmentation mask loss. Calculate $L_{cls}$ and $L_{box}$ and the loss function is calculated in the same way as the Faster R-CNN model, as shown in Eq. (3), because this part is the same as the method used in the Faster R-CNN.

$$L\left(\{p_i\}, \{t_i\}\right) = \frac{1}{N_{cls}} \sum_i L_{cls}\left(p_i, p_i^*\right) + \lambda \frac{1}{N_{\text{reg}}} \sum_i p_i^* L_{\text{reg}}\left(t_i, t_i^*\right) \tag{3}$$

where i is the index value for generating the region of interest: the $N_{cls}$ and $N_{\text{reg}}$ are the number of channels in the feature layer for classification and regression, respectively: and $p_i$ denotes the probability that the first i region of interest is predicted to be the target, and $p_i^* = 0$ When the region is predicted to be a negative sample, the region is predicted to be a positive sample. $p_i^* = 1$ when it is a positive sample, and its two specific calculation procedures are shown in Eqs. (4) and (5):

$$L_{cls}\left(p_i, p_i^*\right) = -\log\left[p_i p_i^* + \left(1 - p_i\right)\left(1 - p_i^*\right)\right] \tag{4}$$
$$L_{reg}\left(t_i, t_i^*\right) = R\left(t_t - t_i^*\right) \tag{5}$$

where $R$ is the SmoothL1 function. The loss function considering the classification and regression can be obtained.

## Model reinforcement of Mask R-CNN models based on attention mechanisms

To harness the complete potential of the feature data gleaned by the backbone feature extraction network and to direct the network's attention toward pertinent image features, including those encompassing spatial location information, this section suggests the incorporation of a module comprising both channel attention and spatial attention within the feature pyramid network (FPN) of Mask R-CNN. The architectural representation of this module is elucidated in Fig. 3.

The channel attention is responsible for discerning inter-channel relationships within the feature maps generated during the feature extraction phase. It accomplishes this by creating channel attention maps, effectively treating each channel as a specialized feature detector honing in on the "what" aspect of the image. Consequently, this enables the network to better apprehend the information contained within channels that pertain to citrus features

(*Woo et al., 2018*). In the course of its operation, the module utilizes mean pooling to consolidate spatial information from the feature map, while maximum pooling excels at preserving unique feature-related information. Therefore, the attention mechanism frequently employs a combination of mean pooling and maximum pooling, leading to a substantial enhancement in network performance. Additionally, computational efficiency is augmented by curtailing the number of channels, rendering the module considerably more resource-efficient.

Generate the required feature vectors for the spatial attention module. The channel attention can be obtained as shown in Eq. (6):

$$
\begin{aligned}
\mathbf{M_c}(\mathbf{F}) &= \sigma(\mathrm{MLP}(\mathrm{AvgPool}(F)) + \mathrm{MLP}(\mathrm{MaxPool}(F))) \\
&= \sigma\left(\mathbf{W}_1\left(\mathbf{W}_0\left(\mathbf{F_{avg}^c}\right)\right) + \mathbf{W}_1\left(\mathbf{W}_0\left(\mathbf{F_{max}^c}\right)\right)\right)
\end{aligned} \tag{6}
$$

where Mc represents the channel attention, $W_0$ isthe feature vector output from the first shared convolutional layer, and $W_1$ is the feature vector output from the second convolutional layer, both have shared weights.

The specific calculation of spatial attention Ms is shown in Eq. (7):

$$
\begin{aligned}
\mathbf{M_s}(\mathbf{F}) &= \sigma\left(f^{7\times7}([\mathrm{AvgPool}(\mathbf{F}); \mathrm{MaxPool}(\mathbf{F})])\right) \\
&= \sigma\left(f^{7\times7}\left(\left[\mathbf{F_{avg}^s}; \mathbf{F_{max}^s}\right]\right)\right).
\end{aligned} \tag{7}
$$

In Eqs. (6) and (7) F represents the feature map of size H*W*C to find the max and average pooling used to represent the weights superimposed in the data. The whole process for the model training is described in Algorithm 1 and the Input data will be detailed described in next section.

## EXPERIMENT

After completing the model construction, we need to train the model. In this section, we first compared the datasets used and screened the relevant data. Then, we used transfer learning methods to complete the training and practical application testing of the model built in this article, and completed the corresponding ablation experiments.

### Datasets

In this manuscript, the research makes use of the PASCAL VOC dataset (https://zenodo.org/records/5095407), which is a widely acknowledged benchmark in the domain of target detection. It is selected for the recognition of certain simpler target instances, specifically focusing on those within the COCO dataset (https://zenodo.org/records/7194300), which currently enjoys broader popularity in the realm of target detection. The COCO dataset stands as a substantial repository, encompassing functionalities related to classification, detection, and segmentation, rendering it one of the most authoritative and publicly accessible datasets in the field of target detection (*Lin et al., 2014*). Comprising natural images as well as everyday objects, this dataset presents intricate backgrounds and often features a multitude of targets within a single image, with an average of 7.7 instance targets and 3.5 distinct categories per picture. Significantly, this dataset poses a formidable

**Algorithm 1: Training process of ATT-MRCNN for the** ___________________

Input: The collected orange picture data, selected COCO and VOC
Initialization.
Define the ATT-MRCNN.
Weights initialization using the COCO datasets with transfer learning manner.
Feature extraction.
Channel attention and Spatial attention mechanism for the feature map.
Model training: Epochs initialization.
while epoch<preset epoch do
Sample data from Input.
Feed data to the ATT-MRCNN.
Model updates.
End
Parameters Fine tuning
while epoch<*preset epoch* do
Validation dataset input.
Loss calculation.
Compute precision, recall and F1-score
Save the optimal model
end
Output: Trained ATT-MRCNN network

challenge, owing to its substantial proportion of small targets. Consequently, nearly all leading models within the field presently employ the COCO dataset for the purposes of evaluation and comparative analysis. The COCO dataset boasts a comprehensive annotation structure, encompassing a total of 80 minor categories (specific item types) like people, bicycles, cars, airplanes, benches, birds, and felines, to name a few. These are further categorized into 12 overarching categories (super-categories) such as furniture, sports, accessories, and outdoor items. In this study, data analysis zeroes in on six specific categories, namely bicycle, car, bus, bird, cat, and dog. For this research, data extraction pertaining to these six categories is executed according to the state-of-the-art category hierarchy method known as "DINO" (*Zhang et al., 2022*). This method streamlines the data selection process for this paper.

The PASCAL VOC dataset, another commonly employed resource in the domain of target detection and segmentation, is also integrated. The training and validation sets of VOC collectively house a total of 11,540 labeled images, with 5,717 in the training set and 5,823 in the validation set. Each image in these sets is accompanied by category labels and rectangular bounding box annotations for every object contained therein. A small subset of objects even includes pixel-level annotations. The dataset encompasses a total of 20 subcategories, encompassing items like people, bicycles, chairs, birds, and cats. These subcategories are further organized into four major categories. In line with the COCO dataset, specific data matching the research's requirements are selected for testing. The specific information for the dataset can be summarized in Table 1 as follows.

**Table 1  The result comparison on the selected COCO dataset.**

| Dataset | Description |
|---|---|
| COCO | The selected type includes: bicycle, car, bus, bird, cat and dog |
| PASCAL VOC | The selected type includes: human, bicycle, chair, bird and cat |

Furthermore, this research constructs its proprietary dataset, aligning with the needs of citrus picking. This dataset predominantly encompasses three categories: No obstruction, Fruit overlapping, and Leaves overlapping, featuring a total of 120 images, collectively serving as a basis for the study.

## Experiment setup and details

Following the dataset creation, the model's evaluation metrics align with the requisites of target detection, focusing predominantly on precision, recall, and F1-score. The calculations for these metrics are detailed in Eqs. (8) through (10). As the public dataset encompasses a broader spectrum of data categories, with the primary emphasis on precision and recall, these metrics are employed to assess the model during both the transfer learning and public dataset testing phases.

$$\text{Precision} = \frac{True\ Positive}{True\ Positive + False\ Positive} \tag{8}$$

$$\text{Recall} = \frac{True\ Positive}{True\ Positive + False\ Negative} \tag{9}$$

$$\text{F1} - \text{score} = \frac{2 * \text{Precision}*\text{Recall}}{\text{Precision} + \text{Recall}} \tag{10}$$

where True Positive indicates the number of citrus targets that were correctly detected, False Positive indicates the number of backgrounds that were detected as citrus targets, and False Negative indicates the number that failed to be detected.

After completing the establishment of the dataset of the model, the evaluation metrics and the associated loss function, the training process of the citrus detection algorithm is shown in Algorithm 1.

## Experiment results and analysis

Upon the successful completion of the training process outlined in Algorithm 1, incorporating the COCO dataset along with the VOC dataset and our in-house dataset, we procured the pertinent results of the model. In the process of model comparison, we subjected the classical Faster R-CNN and the Mask R-CNN, devoid of the integrated channel attention and spatial attention mechanisms, to rigorous data analysis and comparative evaluation. The resultant model metrics derived from the target detection of carefully selected objects within the COCO dataset are meticulously computed and presented in Table 2 and visually represented in Fig. 4.

Based on the recognition data, it is evident that the addition of the attention mechanism yields a substantial enhancement in the detection performance of Mask R-CNN. Moreover, this augmented performance is consistently observed across all six categories of data,
**Table 2 The result comparison on the selected COCO dataset.**

| Method | Faster R-CNN | | | Mask R-CNN | | | Ours | | |
|---|---|---|---|---|---|---|---|---|---|
| Index | Precision | Recall | F1-score | Precision | Recall | F1-score | Precision | Recall | F1-score |
| Bicycle | 0.38 | 0.35 | 0.36 | 0.41 | 0.42 | 0.41 | 0.45 | 0.47 | 0.46 |
| Car | 0.30 | 0.28 | 0.29 | 0.32 | 0.35 | 0.33 | 0.37 | 0.39 | 0.38 |
| Bus | 0.57 | 0.60 | 0.58 | 0.65 | 0.61 | 0.63 | 0.68 | 0.67 | 0.67 |
| Bird | 0.38 | 0.35 | 0.36 | 0.34 | 0.31 | 0.32 | 0.36 | 0.39 | 0.37 |
| Cat | 0.53 | 0.47 | 0.50 | 0.58 | 0.61 | 0.59 | 0.59 | 0.55 | 0.57 |
| Dog | 0.29 | 0.31 | 0.30 | 0.55 | 0.49 | 0.52 | 0.61 | 0.58 | 0.59 |

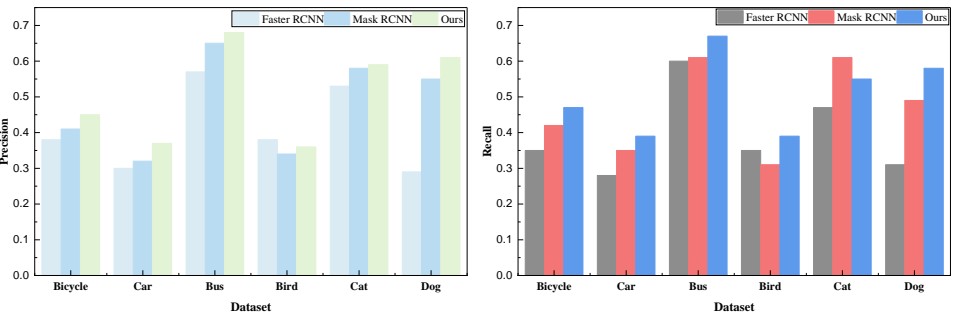

**Figure 4 The result comparison on the selected COCO dataset.**

**Table 3 The result comparison on the selected VOC dataset.**

| Method | Faster R-CNN | | | Mask R-CNN | | | Ours | | |
|---|---|---|---|---|---|---|---|---|---|
| Index | Precision | Recall | F1-score | Precision | Recall | F1-score | Precision | Recall | F1-score |
| Bicycle | 0.46 | 0.44 | 0.45 | 0.53 | 0.55 | 0.54 | 0.55 | 0.57 | 0.56 |
| Car | 0.41 | 0.38 | 0.39 | 0.49 | 0.47 | 0.48 | 0.51 | 0.53 | 0.52 |
| Bus | 0.57 | 0.51 | 0.54 | 0.67 | 0.68 | 0.67 | 0.69 | 0.71 | 0.70 |
| Bird | 0.39 | 0.31 | 0.35 | 0.48 | 0.49 | 0.48 | 0.52 | 0.62 | 0.57 |
| Cat | 0.61 | 0.52 | 0.56 | 0.66 | 0.67 | 0.66 | 0.68 | 0.69 | 0.68 |
| Dog | 0.56 | 0.49 | 0.52 | 0.59 | 0.56 | 0.57 | 0.61 | 0.66 | 0.63 |

surpassing the baseline method. Following the analysis of the COCO dataset, we extended our evaluation to the VOC dataset. The ensuing results, presented in Table 3 and illustrated in Fig. 5, provide further insight:

The data depicted in the figures reveals that the method introduced continues to exhibit commendable performance, especially when tested against the custom-built citrus dataset. In fact, the overall results surpass those achieved with the COCO dataset. This can be attributed to the COCO dataset's complexity and chaotic backgrounds, whereas the VOC dataset comprises more distinct and discernible objects, rendering data recognition a more tractable task.
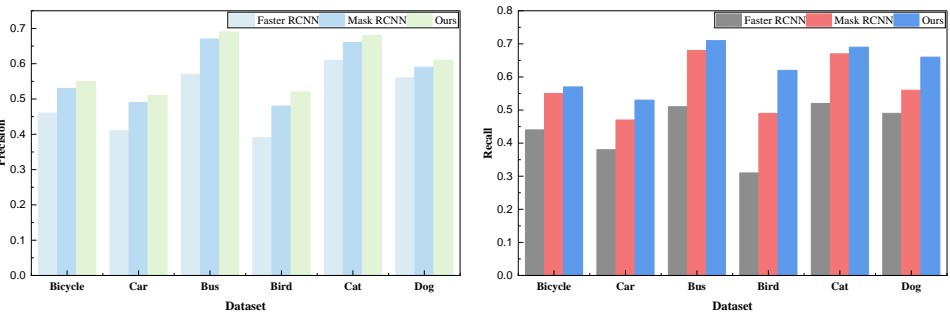

**Figure 5** The result comparison on the selected VOC dataset.

**Table 4** The result comparison on established citrus dataset.

| Method | Faster R-CNN | | | Mask R-CNN | | | Ours | | |
|---|---|---|---|---|---|---|---|---|---|
| Index | Precision | Recall | F1-score | Precision | Recall | F1-score | Precision | Recall | F1-score |
| No obstruction | 0.97 | 0.95 | 0.96 | 0.97 | 0.97 | 0.97 | 0.98 | 0.97 | 0.98 |
| Fruit overlapping | 0.93 | 0.93 | 0.93 | 0.95 | 0.94 | 0.94 | 0.96 | 0.96 | 0.96 |
| Leaves overlapping | 0.93 | 0.91 | 0.92 | 0.94 | 0.94 | 0.94 | 0.95 | 0.93 | 0.94 |

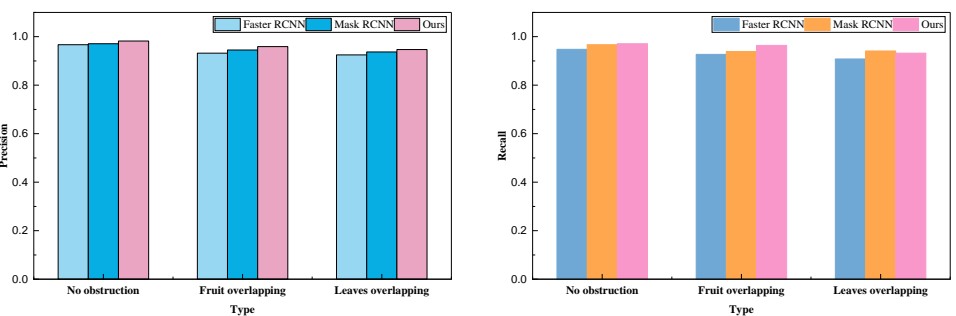

**Figure 6** The result comparison on established citrus dataset.

Subsequent to the analysis of relevant data, we employed the transfer learning approach to conduct model pre-training. Building upon this foundation, we carried out tests on our proprietary citrus dataset, encompassing categories such as "no occlusion," "multi-fruit overlap," and "with branch and leaf occlusion." The data analysis results for these three categories are summarized in Table 4 and Fig. 6.

Transfer learning helps alleviate overfitting problems by transferring features and knowledge learned in the source domain to the target domain, providing better initial states and reducing data requirements in the target domain. This process of sharing knowledge serves as a regularization mechanism, helping the model better generalize to unseen data, especially when data in the target domain is scarce, demonstrating significant advantages. The data analysis clearly demonstrates that the sensory picking model investigated in this

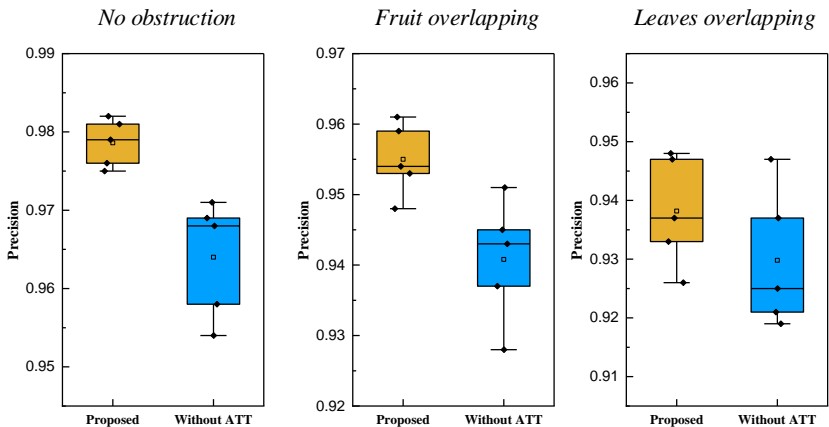

**Figure 7** **The comparison of the established dataset with different batch sizes.**

study, primarily designed for a specific task, attains notably high overall metrics, particularly precision and recall. The recognition rate for the least obstructed type of citrus surpasses 98%, offering valuable reference and technical support for the prospective implementation of intelligent citrus harvesting.

Furthermore, this paper conducted experimentation involving varying batch sizes, and the results of these tests are visualized in Fig. 7.

In Fig. 7, we can see that as the batch size changes, the detection accuracy of ATT-MRCNN for citrus models in different situations does not differ significantly, and overall it is better than models that do not enhance spatial and channel attention. In terms of data and evaluation metrics, the precision of the model has been improved by more than 1% by adding attention mechanisms. Additionally, the incorporation of the attention mechanism yields a stable overall recognition rate for the model proposed, indicating its robustness regardless of changes in batch size. It is not difficult to see through high-precision recognition on the self built database. The successful application of this technology not only improves the efficiency and quality of citrus picking, but also promotes the development of automation and intelligence in the agricultural field. High precision monitoring means that mature and immature citrus fruits can be accurately distinguished, which helps to improve the yield and quality of agricultural production and reduces the labor pressure on farmers. In addition, by reducing reliance on manual labor, deep learning object detection has explored a feasible path for agricultural modernization and provided innovative solutions for sustainable agricultural development.

## DISCUSSION

This article delves into the challenge of citrus image recognition and detection, particularly in the context of intelligent citrus harvesting. It introduces a citrus recognition and detection method founded on deep learning techniques, enhancing the Mask R-CNN model to yield the ATT-MRCNN model, characterized by heightened recognition accuracy. Through rigorous method comparisons involving Faster R-CNN in target detection and the original

Mask R-CNN, the study demonstrates that the addition of channel attention and spatial attention substantially improves recognition accuracy and stability. Given the evolving landscape of computer hardware and edge computing technology, the deployment of such models on compact and lightweight edge devices promises to propel the advancement of intelligent harvesting, significantly contributing to the future of smart agriculture.

Deep learning techniques, with their prowess in computer vision, markedly reduce the demand for human resources and offer solutions to challenges that were hitherto insurmountable through traditional methods. The applications of these techniques in the realms of target detection and intelligent harvesting are extensive, and their potential for further expansion in future research is undeniable. Moreover, the Mask R-CNN model exhibits exceptional transfer learning capabilities, facilitating high-precision application across diverse scenarios with minimal training. As the model deployment advances, it has the potential not only to streamline automatic harvesting but also to refine classification, enabling the detection of citrus ripeness and quality size for more advanced industrial development. Mask R-CNN, augmented with the attention mechanism, is poised to drive more efficient, intelligent, and sustainable agricultural practices, with implications for improved food production, quality, and resource utilization, while concurrently alleviating the need for manual labor. This presents a profound opportunity for the future of agriculture, addressing the challenges of global food production and sustainable development. Object detection technology has a wide range of applications beyond the field of agricultural harvesting. In industrial automation, it can be used for quality inspection and object tracking on production lines, improving production efficiency and product quality. In the field of intelligent transportation, object detection technology can be used for real-time traffic monitoring, vehicle and pedestrian recognition, which helps to improve the level of traffic safety management. In addition, disease diagnosis in the medical field, obstacle detection in autonomous vehicles, and abnormal behavior detection in security monitoring systems are also important application directions of object detection technology, demonstrating its diversified potential in different fields.

## CONCLUSION

This study innovatively proposes an intelligent citrus object detection model based on ATT-MRCNN, aiming to provide strong support for the improvement of intelligent citrus picking algorithms. Through simple testing and pre training on the gold standard datasets for object detection such as COCO and VOC, we designed classification object detection tasks for three types of data: unobstructed, multi fruit overlapping, and branch and leaf occlusion according to actual needs. The experimental results show that compared to the Mask R-CNN method without added attention mechanism and the basic Faster R-CNN method, the method proposed in this paper exhibits better performance. Meanwhile, the test results on public datasets such as COCO also show significant advantages. In actual citrus picking data model testing, this method achieved a recognition precision of over 98% for unobstructed citrus targets, significantly better than the 97% without added attention mechanism, thus significantly improving the recognition accuracy and stability of the

model. The model framework of this innovative method provides useful references for future research on intelligent harvesting and cultivation in agriculture, and has important driving significance for the intelligence and modernization of agricultural development.

In forthcoming research endeavors, the aspiration is to enhance the model's generalization capabilities. Additionally, extending the analysis to encompass different leaves and the age of citrus trees promises to furnish deeper insights for the subsequent analysis of citrus data. Despite the ATT-MRCNN model's commendable target detection performance, further exploration is warranted to assess its real-time detection capabilities and performance under varying weather conditions.

## ACKNOWLEDGEMENTS

We thank the anonymous reviewers whose comments and suggestions helped to improve the manuscript.

### Funding
The study received no funding.

### Competing Interests
The authors declare there are no competing interests.

### Author Contributions
- Ziwei Guo conceived and designed the experiments, performed the experiments, prepared figures and/or tables, authored or reviewed drafts of the article, and approved the final draft.
- Yuanwu Shi conceived and designed the experiments, performed the experiments, analyzed the data, performed the computation work, prepared figures and/or tables, authored or reviewed drafts of the article, and approved the final draft.
- Ibrar Ahmad analyzed the data, performed the computation work, prepared figures and/or tables, authored or reviewed drafts of the article, and approved the final draft.

### Data Availability
The code is available in the Supplemental Files.

The data is available at Zenodo:

- Giuseppe Amato, Paolo Bolettieri, Fabio Carrara, Fabrizio Falchi, Claudio Gennaro, Nicola Messina, Lucia Vadicamo, & Claudio Vairo. (2022). COCO, LVIS, Open Images V4 classes mapping [Data set]. Zenodo. https://doi.org/10.5281/zenodo.7194300

- Lemmens, J. (2021). Object Detection meets Knowledge Graphs Raw Data [Data set]. Zenodo. https://doi.org/10.5281/zenodo.5095407

## Supplemental Information

Supplemental information for this article can be found online at http://dx.doi.org/10.7717/peerj-cs.1865#supplemental-information.

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
