# Peer review of "Design of smart citrus picking model based on Mask RCNN and adaptive threshold segmentation"

_PeerJ Computer Science, doi:10.7717/peerj-cs.1865_

## Round 0.1 · original submission · Major Revisions

The authors should revise as per reviewer comments.

**Language Note:** PeerJ staff have identified that the English language needs to be improved. When you prepare your next revision, please either (i) have a colleague who is proficient in English and familiar with the subject matter review your manuscript, or (ii) contact a professional editing service to review your manuscript. PeerJ can provide language editing services - you can contact us at copyediting@peerj.com for pricing (be sure to provide your manuscript number and title). – PeerJ Staff

·

Basic reporting

Paper is reversed well

Experimental design

Experiment design is well

Validity of the findings

Validity is valid

Additional comments

Paper is well revised and accepted

Reviewer 2 ·

Basic reporting

In this article, the Expo delved into automated citrus harvesting and authors proposed an ATT-MRCNN object detection model that seamlessly integrates channel attention and spatial attention mechanisms for recognizing citrus images, achieving recognition rates above the 95% threshold across three sensory recognition tasks. However, this paper still has shortcomings that are listed below:

- In the related works, the author can consider introducing from the application background, and then reference many relevant models to compare, and then expand the introduction;
- In the third part of the model introduction, some formulas lack the relevant formula description introduction and explanation, which is not easy for readers to understand;
- The introduction to the data set in section 4.1 is a bit confusing with a large section of text, and it is more convenient for readers to read and understand it by changing it to a chart form;

Experimental design

- The part of experimental results lacks the description of the relevant specific values, and the practical significance of the experimental results is not expressed, which needs to be strengthened in the following two aspects;

Validity of the findings

- As for the specific contribution, data extraction and model pre-training of the two data sets by using the model lacks innovation, and the author needs to consider strengthening innovation points;

Additional comments

- In section 4.3, the attention mechanism adds detection performance to the algorithmic model in this paper, in terms of how much correlation performance is enhanced?
- Considering the discussion section, it seems that the overall content is too small to support the framework of the discussion. It can be described more from multiple aspects;
- The language of the whole article may be slightly inadequate, such as in grammar and professionalism, which needs to be improved.

·

Basic reporting

This paper introduces a smart citrus target detection model based on ATT-MRCNN, aiming to strengthen the algorithm support for smart citrus harvest. Empirical tests on actual citrus harvest data show that the accuracy of the re-design of accessible citrus targets is more than 98%, but the following parts still need to be improved:
1. The title is not appropriate, and I would suggest that the author change it to some kind of enhanced model based on some kind of algorithm for smarter picking;
2. Each section needs to add a first paragraph to summarize and explain the general content of each section to improve the readability.

Experimental design

3. What are the specific functions of the module of Mask RCNN model and RolAlign layer in section 3.2, which the author can elaborate in detail?
4. As mentioned in section 4.1, this study constructed a proprietary dataset with 120 images in three categories, whether too few image datasets would result in overfitting;
5. The training process of the citrus detection algorithm in Section 4.2 should not be included in this section, but in the third part of the Model and introduction.

Validity of the findings

6. In the experiment, the author used transfer learning to pre-train the model. What effect did the pre-training have on the subsequent model training?
7. The content of the conclusion is quite different from that of the abstract, and the author should strengthen the description of the conclusion from other aspects;
8. Most of the references are not in recent years. I would suggest adding some excellent journal articles in recent years to enhance the timeliness of the articles.

---

## Round 0.2 · accepted · Accept

The authors have revised the article based on the reviewers' comments.

Reviewer 2 ·

Basic reporting

All the suggested changes have been completed

Experimental design

All the suggested changes have been completed

Validity of the findings

All the suggested changes have been completed

Additional comments

All the suggested changes have been completed

·

Basic reporting

The authors has incorporated my previous suggestions in current version. Therefore it is fine now

Experimental design

The experimental design has been improved well in the revised paper

Validity of the findings

The findings of the paper are valid and can be helpful for the community related to the agriculture field.

Additional comments

Overall the quality of the paper is well and it is recommended to publish in it's current form.